# Prevalence of and Factors Associated with Depressive Symptoms among Indonesian Migrant Workers in Taiwan

**DOI:** 10.3390/ijerph20054056

**Published:** 2023-02-24

**Authors:** Yuni Asri, Kun-Yang Chuang

**Affiliations:** 1School of Public Health, Taipei Medical University, 250 Wu-Hsing Street, Taipei 11031, Taiwan; 2Department of Nursing, Institut Teknologi, Sains dan Kesehatan RS dr Soepraoen Kesdam V/Brawijaya, Malang 65147, Jawa Timur, Indonesia

**Keywords:** mental health, depressive symptoms, migrant workers, Indonesian, Taiwan

## Abstract

International migrant workers contribute significantly to the economic growth of the receiving country, and yet their health, especially their mental health, has long been overlooked. The purpose of this study was to identify the factors associated with depressive symptoms among Indonesian migrant workers in Taiwan. This study used cross-sectional data from 1031 Indonesian migrant workers in Taiwan. Demographic, health, and living- and work-related variables, as well as depressive symptom variables assessed using the Center for Epidemiological Study on depressive symptoms scale, were collected. Logistic regression analysis was used to identify related factors. About 15% of the Indonesian migrant workers had depressive symptoms. The significant factors associated with these symptoms were age, educational level, frequency of contact with families, self-rated health, time spent in Taiwan, region of work, satisfaction with the living environment, and freedom to go out after work. The findings thus identify target groups who are more likely than others to suffer from depressive symptoms, and we suggest appropriate approaches for devising interventions to reduce depressive symptoms. The findings of this research suggest the need for targeted approaches to reducing depressive symptoms among this population group.

## 1. Introduction

One of the most important social phenomena in the global context is the flow of international migrant workers. It was estimated that there were about 169 million international migrant workers (IMWs) worldwide in 2019 [1]. International migrant workers face health issues similar to those of underserved populations. Furthermore, their problems are often magnified or compounded by their status and by their need for cultural adjustment. International migrants struggle with transportation, language, and cultural differences, lack familiarity with local healthcare services, and have limited eligibility for publicly and privately funded healthcare programs [2]. Most IMWs come from lower-income countries [3,4] and the vast majority of them work in manufacturing, construction, and agricultural industries, as well as in domestic services [4,5]. They provide a steady supply of labor and play a vital part in the economic growth of upper-income countries [6,7]. Despite its importance, the health and wellbeing of IMWs is frequently overlooked. Furthermore, IMWs’ problems are often compounded by their status. Many of them deal with hazardous working and living conditions, overwhelming work stress, loneliness, and prejudice [8,9,10,11,12,13]. Moreover, social adjustment, language barriers, and cultural factors also significantly impede their access to care [2]. All these factors have a negative effect on their wellbeing [14,15]. Researchers in the US, Europe, and Asia have found that temporary international migrant workers experience mental health challenges [16]. Previous studies have indicated a higher prevalence of depressive symptoms among migrant workers than in the general population [17,18]. The factors associated with depression among international migrant workers include poor working conditions [16,18], poor relationships with employers [19], poor nutrition [20], insufficient sleep [21], and abuse [11,16]. These factors vary depending on the destination country. While much research concerning the wellbeing of IMWs has focused on aspects of physical health and the use of services [10,14,15,22,23,24], little research has been conducted on the mental health of IMWs. 

Recent research on IMWs’ mental health has shown an alarmingly high percentage of migrant workers with mental health issues. The most commonly reported mental health problems were depressive symptoms and anxiety disorders [25,26]. The prevalence of IMWs showing signs of depressive symptoms has been reported to be as high as 20% [17,27].

Globally, about 20.4% of IMWs originate from Asia and the Pacific regions. Indonesia is a major source of migrants in the Asia-Pacific region, and Taiwan is a major destination for IMWs from Indonesia. There were more than 700,000 IMWs from Southeast Asia working in Taiwan in the end of 2020, of whom 37.1% were from Indonesia, 33.4% were from Vietnam, 21.3% were from the Philippines, and 8.2% were from Thailand and other countries [28]. IMWs work in various sectors of industry, and include construction workers, domestic helpers, factory workers, and farm and fishery workers [29]. Taiwan is a major destination for international migrant workers in Asia, while Indonesia is the one of the most populous nations and a major source of migrants in the Asia-Pacific region [30]. Hence, the aim of this study was to estimate the prevalence of depressive symptoms among Indonesian IMWs in Taiwan and to identify factors that may be associated with depressive symptoms. The findings of this research can be used to improve the wellbeing of IMWs in Taiwan and elsewhere.

## 2. Materials and Methods

### 2.1. Study Design and Setting

This is a cross-sectional study, which was conducted in Taiwan from April to August 2019. Participants were recruited using convenience sampling. One of the major difficulties in migrant health research is the recruitment of a sufficiently large number of diversified subjects. Time and sample size concerns were the reasons for recruiting Indonesian migrant workers in Taiwan. The inclusion criteria were that the participants were Indonesian migrant workers who were willing to finish the survey, were aged ≥ 18 years, and had been living in Taiwan for at least 6 months. The required sample size for this study was 384 Indonesian migrant workers, and the sample size in this study was 1031. This was sufficient to survey a population of Indonesian migrant workers of around 263,358 in Taiwan with a 95% confidence level and 5% margin of error.

Because of the lack of an appropriate sampling frame and means of contacting participants, the participants were recruited through social media and websites that target migrant workers. The participants were invited to fill out an online questionnaire. The researchers traveled to major cities in the northern, central, southern, and eastern regions of Taiwan to recruit participants. IMWs in Taiwan have a unique social pattern. At weekends, they tend to gather at specific locations such as train stations, shopping malls, or mosques. On weekdays, it is more likely that they will be found in community parks or at night markets. 

Moreover, at various locations where migrant workers gather, the researchers also handed out information regarding this research, online QR code stickers, and website information to migrant workers, and invited migrant workers to participate in this research. The data collection methods allowed migrant workers to respond to the questionnaire in private, when it was convenient for them to do so. The website could be accessed using either a computer, a pad, or a smart phone. It took about 10~15 min to complete the questionnaire. If a person was unable to fill out the questionnaire at the time, they were given an online QR code sticker so they could complete it online later. This allowed migrant workers to respond to the questionnaire in private if they wanted to. The questionnaires did not elicit personal identifying information from the participants, and the data could not be traced back to the individual participants. At the end of the data collection, any submitted questionnaires with identical IP addresses or emails were deleted to minimize duplicate entries. The number of participants in this study was 1038 people; after deleting respondents with missing data, a total of 1031 Indonesian migrant workers had completed the questionnaire and were included in the analysis. This study was approved by the Taipei Medical University Ethics Committee (N201903024).

### 2.2. Measurements

Among the various tools used, depression was assessed according to the Center for Epidemiologic Studies Depression Scale (CES-D) [31]. In primary care settings, a common tool for detecting depression is the CES-D-10. In community populations, this 10-item measure has been shown to have robust psychometric qualities, such as high correlations with the original 20-item form, and excellent predictive accuracy [32]. The 10-item version has been previously used in an Indonesian study [33]. The CES-D 10 items question had four response options: rarely or none of the time (less than 1 day), some or a little of the time (1–2 days), occasionally or a moderate amount of time (3–4 days), and most or all of the time (5–7 days). The total scores were calculated as the unweighted sum of the 10 component items, with a potential range of 0~30, and a cutoff score of ≥10 was considered to indicate depressive symptoms [34].

Demographic, family, health, and social and living characteristics were included in the analysis as predictors. The demographic variables included sex (male or female), age (≤24, 25~29, 30~34, or ≥35 years), educational level (junior high school or lower; or senior high school or higher), and marital status (unmarried or married). The family characteristics included whether the participants had children (yes or no), the frequency of contact with family members (≥twice a week or <twice a month), and whether they were supporting their family at home financially (yes or no). The health variables included self-rated health and unmet needs. Self-rated health was assessed with one question: “How good is your health?” The response categories “very good”, “good”, and “average” were grouped together to represent “good health”, while the categories “poor” and “very poor” were grouped together to represent “poor health”. Unmet needs were assessed with the question “While in Taiwan, was there ever a time that you felt you needed medical help (examination or treatment) but you did not receive it?” (yes or no).

The social and living characteristics included time spent in Taiwan (<2 or ≥2 years), the region in which the migrants worked (northern, central, southern, or eastern), and number of working days per week (6 days or fewer, or 7 days). The satisfaction with the living environment variable was assessed using one question: “In general, are you satisfied with your personal living environment in Taiwan?” (satisfied or unsatisfied). The freedom to go out was assessed by asking “Are you allowed to go out on your own in your free time?” (yes or no), and the amount of leisure activities was assessed by asking “To what extent do you have the opportunity for leisure activities?” (little or less; or moderate or more). The questionnaire was first developed in English and was then translated into Bahasa Indonesia. Back translation was carried out to ensure accuracy.

### 2.3. Statistical Analysis

The data were described using frequency (n) and percentage (%). Bivariate analysis of the respondents’ characteristics and depressive symptoms was conducted with a chi-squared test to compare the categorical variables: all the tests were bilateral at *p* < 0.05. Multivariate logistic regression analysis was used to identify the factors associated with depression among the Indonesian migrant workers in Taiwan, and all the variables were considered for inclusion in the adjusted model. Odds ratios (ORs) and 95% confidence intervals (CIs) were generated for each variable in the final model. The significance level was set at *p* < 0.05. Data analysis was performed using the Statistical Package for Social Science (SPSS) software version 25.0 for windows (IBM Corp., Armonk, NY, USA).

## 3. Results

In total, 1039 Indonesian migrant workers filled out the questionnaire. After excluding the cases with missing values, 1031 were included in the final analysis. Table 1 shows the basic characteristics of the study sample. Of the sample, 53.4% were females, 68.1% were older than 30 years, most (89.2%) had an education level of senior high school or above, 65.4% were married, and 76.5% had children in their home country. As many as 92.5% of the migrant workers were financially supporting family members in their home country, and more than 69% of them had contact with family members more than twice a week. In terms of health, 53.9% had poor self-rated health, and 42.3% had unmet medical needs. Among the workers, 15% reported a CES-D score of >10, and hence were categorized as having depressive symptoms. 

Table 2 shows the results of our assessment of the associations between depressive symptoms and predictor variables. With the exception of educational level and unmet medical needs, all the other variables were significantly associated with depressive symptoms. Being female, being younger, being unmarried, having no children, having fewer contacts with one’s family, having less financial pressure, being in poor health, having spent <2 years in Taiwan, working in the northern region, working more days per week, being unsatisfied with one’s living environment, being unable to go out when not working, and having less time for leisure activities were associated with depressive symptoms.

Table 3 shows the factors associated with depressive symptoms according to the multivariate logistic analyses. After controlling for other variables, those who were more than 30 years old, who were more highly educated, who had more frequent contact with their families, who had better health, and who had been in Taiwan longer were less likely to have depressive symptoms than their counterparts. Furthermore, those working in the central, southern, or eastern regions were less likely to have depressive symptoms. Satisfaction with one’s living environment and the ability to go out when not working were also associated with fewer depressive symptoms.

## 4. Discussion

Depressive symptoms are a common mental health problem among migrant workers. As high as 15% of the Indonesian migrant workers who took part in this study showed signs of depressive symptoms. This prevalence is similar to that found in previous studies using the same CES-D scale. In a study in Saudi Arabia, the prevalence was 20% [17], in Mexican immigrants to the United States (US) it was approximately 16% [35] and among Turkish migrants in the Netherlands it was 17.1% [36].

This study showed that being older was associated with a lower likelihood of showing depressive symptoms, even after controlling for family characteristics and working years. Other studies also identified that age may be a protective factor against depressive symptoms among migrant workers [17], although it is not clear why older migrants are less likely to have depressive symptoms. There has been some speculation that older migrants are better adapted to a migratory lifestyle, are more experienced, and are more accustomed to utilizing local resources. However, more research is needed to explain why age was a significant factor, even after controlling for other work-related and health variables. Education was another demographic characteristic associated with depressive symptoms. Consistent to other studies [36,37], this research found that migrant workers with a higher education level were less likely to show depressive symptoms, probably because they have better language skills and more knowledge of how to utilize formal resources specifically for migrant workers. Hence, based on the findings of this research, government policies targeting migrant workers’ health should place more emphasis on those who are younger and less educated.

In terms of family-related factors, some previous research has identified financial burdens and concerns for children as being associated with depressive symptoms among migrant workers [38]. However, in this research, financial and children’s factors were not significantly associated with depressive symptoms. This can probably be explained by the significant differences in salary and living expenses between Indonesia and Taiwan. The average salary for a migrant manual laborer in Taiwan is about USD735.46 [39], much higher than the average salary of around USD198.95 in Indonesia [40]. Hence, it would seem that this salary is sufficient for migrants to provide for their families back home. A low frequency of contact with families, either by telephone or via the internet, was significantly associated with depressive symptoms. Hence, it is essential to ensure that migrant workers have the means, such as privacy, free time, and low-cost internet access, to maintain regular contact with their family back home.

In terms of social and living characteristics, having been in Taiwan longer, being more satisfied with the living environment, being able to go out freely, and working outside of the Taipei area were associated with fewer depressive symptoms. This is consistent with the results of a previous study, in which migrant workers who had been in Taiwan longer were less likely to have depressive symptoms [41]. Again, this finding highlights the need for policy interventions to assist migrant workers who have just arrived. Dissatisfaction with the living environment and the inability to go out freely were associated with depressive symptoms. Migrant workers in the manufacturing and construction industries usually stay in dormitories maintained by their employers. Even though there is an official regulation regarding minimum dormitory standards, such as a minimum of 3.2 m^2^/person [42], this regulation is probably insufficient to ensure the quality of the living environment.

Another factor associated with depressive symptoms was the freedom to go out when not working. Some migrant workers, especially domestic helpers, are caregivers of a disabled person, a job that leaves them with little free time, even at weekends. There are sporadic reports on the abuse of migrant workers who are being asked to work seven days a week, sometimes without extra pay. Thus far, the government has been reluctant to strictly enforce a six-day working week, but instead allowing employers to request workers to work on their days off for additional payment. However, migrant workers are not usually in a position to deny employers’ requests. To reduce the abuse of migrant workers, governments should scale up prevention and intervention measures. Some measures that have been proved to be successful in reducing migrant labor abuse are the implementation of clearly written contracts, and the enforcement of health and safety regulations [11,43].

Migrant workers in northern Taiwan had a higher prevalence of depressive symptoms than workers in other regions, despite the fact that northern Taiwan has the highest standard of living, a comprehensive public transportation system, and many international residents and visitors. While all these characteristics are reasons for choosing a place to live among expatriates [44], living in northern Taiwan could also mean that the living environment is more crowded, with smaller personal spaces, a higher cost of living, greater work pressures, and less personal freedom. These factors may explain the higher likelihood of depressive symptoms among migrant workers in northern Taiwan.

This is the first study to focus on the mental health of international migrant workers in Taiwan. However, it has several limitations. This study utilized a cross-sectional study design, and the results should be interpreted accordingly. Caution is also needed when generalizing the results to other migrant workers from other countries, since cultural differences, as well as the size of the social network, may influence the results. Since this study did not employ probability sampling and used an online questionnaire to collect the data, the identity of the participants could not be verified. However, the use of convenience sampling did allow us to have a larger study sample than previous studies. Nevertheless, migrant workers who did not have internet access were excluded, especially those working in remote and isolated areas, such as in the agricultural or fishery industries. Hence, it is likely that the prevalence of depressive symptoms has been underestimated. For future research, it may be beneficial to consider other sampling methods, such as random sampling, to ensure a more representative sample.

This research provides an estimate of the prevalence of depressive symptoms among Indonesian migrant laborers in Taiwan, which was around 15%. In addition to other previously identified factors, the frequency of contact with families in migrants’ home countries, the living environment, and the ability to go out when not working were associated with depressive symptoms. Interventions to improve mental health among migrant workers in Taiwan can begin by improving the living environment and by enforcing regulations protecting the rights of migrant workers.

## 5. Conclusions

In summary, about 15% of Indonesian migrant workers who participated in the study had depressive symptoms, and the variables of age, educational level, frequency of contact with families, self-rated health, time spent in Taiwan, region of work, satisfaction with the living environment, and the ability to go out when not working were associated with the depressive symptoms International migrant workers contribute significantly to the economic development of the receiving country, as well as providing a source of income to alleviate poverty in their home countries. However, they face an array of challenges that may be detrimental to their mental health. The prevalence of depressive symptoms among international migrant workers is high. Given the migrants’ disadvantaged status, governments and policymakers have allocated few resources to improving their working and living conditions, and have made little infrastructural changes to improve their health. This research has identified those migrants who are more likely than others to suffer from depressive symptoms. Hence, interventions by the authorities should focus on target populations. Moreover, the findings of this research indicate that to ameliorate the problem of depressive symptoms among migrant workers, governments should scale up efforts to improve migrants’ living and working environments. The findings also suggest the need for targeted approaches for reducing depressive symptoms among this population group. Regulations concerning living quarters and working hours should be strictly enforced. Poor mental health among international migrant workers could lead to low productivity and economic losses, and hence efforts and resources aimed at improving the health of international migrant workers could prove to be beneficial to economic growth and social stability.

## Figures and Tables

**Table 1 ijerph-20-04056-t001:** Frequency distribution of characteristics among Indonesian migrant workers (n = 1031).

Variable	Frequency (n)	Percent (%)
*Demographic characteristics*		
Sex		
Male	480	46.6
Female	551	53.4
Age group		
≤24 years	124	12.0
25~29 years	205	19.9
30~34 years	246	23.9
≥35 years	456	44.2
Educational level		
Junior high school or lower	111	10.8
Senior high school or higher	920	89.2
Marital status		
Unmarried	357	34.6
Married	674	65.4
*Family characteristics*		
Have children		
Yes	789	76.5
No	242	23.5
Frequency of contact with family members		
≥Twice a week	713	69.2
<Twice a month	318	30.8
Support family back home financially		
Yes	954	92.5
No	77	7.5
*Health*-*related variables*		
Self-rated health		
Poor	556	53.9
Good	475	46.1
Unmet medical needs		
Yes	436	42.3
No	595	57.7
*Social and living characteristics*		
Time spent in Taiwan		
<2 years	101	9.3
≥2 years	930	90.2
Region of work		
Northern	308	29.9
Central	296	28.7
Southern	306	29.7
Eastern	121	11.7
Number of working days per week		
6 days or fewer	482	46.8
7 days	549	53.2
Satisfied with living environment		
Satisfied	810	78.6
Unsatisfied	221	21.4
Able to go out when not working		
Yes	665	64.5
No	366	35.5
Opportunity for leisure activities		
Little or less	633	61.4
Moderate or more	398	38.6

**Table 2 ijerph-20-04056-t002:** Chi-squared analysis of factors associated with depressive symptoms among Indonesian migrant workers in Taiwan.

Variable	Without Depressive Symptoms		With Depressive Symptoms		*p* Value
	N	%	N	%	
*Demographic characteristics*					
Sex *					0.000
Male	454	94.6	26	5.4	
Female	422	76.6	129	23.4	
Age group *					0.000
≤24 years	73	58.9	51	41.1	
25~29 years	143	69.8	62	30.2	
30~34 years	222	90.2	24	9.8	
≥35 years	438	96.1	18	3.9	
Educational level					0.675
Junior high school or lower	93	83.8	18	16.2	
Senior high school or higher	783	85.1	137	14.9	
Marital status *					0.000
Unmarried	262	73.4	95	26.6	
Married	614	91.1	60	8.9	
*Family characteristics*					
Have children *					0.000
Yes	709	89.9	80	10.1	
No	167	69.0	75	31.0	
Frequency of contact with family members *					0.000
≥Twice a week	676	94.8	37	5.2	
<Twice a month	200	62.9	118	37.1	
Support family back home financially *					0.004
Yes	820	86.0	134	14.0	
No	56	72.7	21	27.3	
*Health-related variables*					
Self-rated health *					0.000
Poor	409	73.6	147	26.4	
Good	467	98.3	8	1.7	
Unmet medical needs					0.052
Yes	359	82.3	77	17.7	
No	517	86.9	78	13.1	
*Social and living characteristics*					
Time spent in Taiwan *					0.000
<2 years	34	33.7	67	66.3	
≥2 years	842	90.5	88	9.5	
Region of work *					0.000
Northern	234	76.0	74	24.0	
Central	258	87.2	38	12.8	
Southern	272	88.9	34	11.1	
Eastern	112	92.6	9	7.4	
Number of working days per week *					0.000
6 days or fewer	455	94.4	27	5.6	
7 days	421	76.7	128	23.3	
Satisfied with living environment *					0.000
Satisfied	764	94.3	46	5.7	
Unsatisfied	112	50.7	109	49.3	
Able to go out when not working *					0.000
Yes	632	95.0	33	5.0	
No	244	66.7	122	33.3	
Opportunity for leisure activities *					0.000
Little or none	497	78.5	136	21.5	
Moderate or more	379	95.2	19	4.8	

Note: * *p* < 0.05.

**Table 3 ijerph-20-04056-t003:** Logistic regression of factors associated with depressive symptoms among Indonesian migrant workers in Taiwan.

Variable	Odds Ratio (OR)	95% Confidence Interval (CI)
*Demographic characteristics*		
Sex		
Male	1.00	-
Female	2.50	0.28~13.13
Age group		
≤24 years	1.00	-
25~29 years	0.70	0.29~1.69
30~34 years	0.30 *	0.09~0.95
≥35 years	0.24 *	0.07~0.81
Educational level		
Junior high school or lower	1.00	-
Senior high school or higher	0.24 *	0.09~0.59
Marital status		
Unmarried	1.00	-
Married	1.22	0.59~2.51
*Family characteristics*		
Have children		
Yes	1.00	-
No	0.48	0.20~1.14
Frequency of contact with family members		
≥Twice a week	1.00	-
<Twice a month	5.28 *	2.49~11.21
Support family back home financially		
Yes	1.00	-
No	0.61	0.26~1.44
*Health-related variables*		
Self-rated health		
Poor	1.00	-
Good	0.23 *	0.09~0.55
Unmet medical *needs*		
Yes	1.00	-
No	0.62	0.35~1.08
*Social and living characteristics*		
Time *spent* in Taiwan		
<2 years	1.00	-
≥2 years	0.13*	0.06~0.28
Region of work		
Northern	1.00	-
Central	0.23 *	0.11~0.49
Southern	0.24 *	0.12~0.48
Eastern	0.09 *	0.03~0.26
Number of days worked per week		
6 days or fewer	1.00	-
7 days	1.06	0.21~5.34
Satisfied with living environment		
Satisfied	1.00	-
Unsatisfied	5.20 *	2.74~9.89
Able to go out when not working		
Yes	1.00	-
No	2.42 *	1.01~5.84
Opportunity for leisure activities		
Little or less	1.00	-
Moderate or more	1.32	0.59~2.92

Note: * *p* < 0.05.

## Data Availability

The datasets generated during and/or analyzed during the current study are available from the corresponding author on reasonable request.

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
