# Peer review of "Prevalence of and Factors Associated with Depressive Symptoms among Indonesian Migrant Workers in Taiwan"

_ijerph, 2023, doi:10.3390/ijerph20054056_

Round 1
Reviewer 1 Report
Dear Authors,
Thank you for submitting your work. This is very interesting and provides relevant data to the field.
I would like to suggest some points in order to help improve the quality of your paper.
Major concerns
Introduction
I suggest that you can expand this section in order to look at the prevalence of depression symptoms globally.
Additionally, explain the scientific background and rationale for the investigation being reported.
Materials and Methods
Explain how the study size was arrived at. I propose that a clear indication must be given of how a representative sample size of the study was calculated.
Clearly define all outcomes, exposures, predictors, potential confounders, and effect modifiers. Give diagnostic criteria, if applicable.
Please replace the term “depression“ to “depressive symptoms“ throughout this manuscript.
Line 79: Please replace the term „gender (male or female)“ to „sex (male or female)“ throughout the paper.
Describe any efforts to address potential sources of bias.
Describe all statistical methods, including those used to control for confounding.
Describe any methods used to examine subgroups and interactions.
Explain how missing data were addressed.
Results
Table 3. The eligibility criterion for the logistic regression model seems to be required (Nagelkerke R Squared). Nagelkerke R Squared is an adjusted version of Cox and Snell R Squared. It measures the proportion of the total variation of the dependent variable can be explained by independent variables in the current model.
Indicators such as beta values (with standard error) must be provided as well as Wald statistics in logistic regression too.
Discussion
Well-written.
Conclusions
It is necessary to clarify conclusions according to the aim of this study. The aim of this study was to estimate the prevalence of depressive symptoms among Indonesian IMW in Taiwan and to identify factors that may be associated with depression.
Minor concerns
Line 247: What was the date of issue of the Ethics Committee permission?
Warm Regards
Reviewer 2 Report
In this study, Asri and Chuang aimed at identifying factors associated with depression among Indonesian migrant workers in Taiwan. They found that 15% of workers had depressive symptoms from a sample of 1031 workers and found that factors such as age, education, family contact, self-rated health, length of time in Taiwan, work location, living satisfaction and free time were related to depression. The findings suggest the need for targeted approaches for reducing depression among this population group.
Thank you for the opportunity to review this manuscript, which I found to be a valuable contribution to this crucial issue. The study extends previous similar observations made in different countries and can be a useful starting point for more structured research and establishing new policies to reduce the burden of psychiatric symptoms in immigrant workers.
I have some comments and suggestions regarding the methodology, results, and presentation of the paper, that I believe could enhance the quality of the article.
- The cross-sectional design of the study is appropriate for the research question and provides valuable information about the study population at a single point in time. The use of a convenience sampling approach is also understandable, given the lack of a sampling frame or contact information, even though it may introduce a selection bias, and the authors are aware of it as stated in the study limitations. Nevertheless, the approach of recruiting participants in different regions of Taiwan is commendable to help ensure that the study is representative of the study population. For future research, it may be beneficial to consider other sampling methods, such as stratified or random sampling, to ensure a more representative sample.
- I recommend providing a short description of the 10-item version of the CES-D and explaining how it differs from the 20-item version.
- There is no information about the characteristics of the participants who were approached but declined to participate in the study, which could limit the interpretation of the results. The authors should provide more information (at least declare how many persons were approached).
- Report that you use the Fisher exact test p-value for the Chi-square test in Table 2 and report the statistic, not only the p-value.
- The paper would greatly benefit from a review to eliminate duplicated words, typos, and grammatical errors to ensure that the intended meaning is effectively conveyed.
For example, on line 24 there is a missing space between “workers (IMW)” and “worldwide”, on line 38 “Recent researched” should be “recent research”. Additionally, on line 40: “ Depression” should be in lowercase, but there are more.
Overall, I recommend this manuscript for publication once these minor revisions have been addressed.
Round 2
Reviewer 1 Report
The Authors answered my questions.
Kind Regards